# Supported Eosin Y as a Photocatalyst for C-H Arylation of Furan in Batch and Flow

**DOI:** 10.3390/molecules27165096

**Published:** 2022-08-10

**Authors:** Sergio Rossi, Fabian Herbrik, Simonetta Resta, Alessandra Puglisi

**Affiliations:** Dipartimento di Chimica, Università degli Studi di Milano, Via Golgi 19, 20133 Milano, Italy

**Keywords:** photocatalysis, eosin Y, supported catalysis, continuous flow photochemistry, green chemistry

## Abstract

Eosin Y is one of the most popular organic dyes used as a photoredox catalyst and is largely employed in photochemical reactions both as a homogeneous and heterogeneous photocatalyst after immobilization. Immobilization of Eosin Y onto a solid support has many advantages, such as the possibility of recovery and reuse of the photocatalyst and the possibility of its use under flow conditions. In this paper, we report our findings on the immobilization of Eosin Y onto Merrifield resin and its application in the direct photochemical arylation of furan with aryldiazonium salts. The synthesized supported photocatalyst was used in batch reactions under heterogeneous conditions with different aryl diazonium salts, and its recovery and recycle were demonstrated for up to three times. The immobilized photocatalyst was then loaded in a packed-bed reactor and used under continuous flow conditions. The flow reaction allowed the arylated products to be obtained with higher productivity and space-time-yield than the batch in a very short reaction time.

## 1. Introduction

In 1912, Professor G. Ciamician envisioned a future in which chemical processes would be mostly run by a sustainable, abundant, and never-ending energy source: photons coming from the sun [1,2]. In his 2008 famed study, David MacMillan turned the spotlight back on photochemistry, more specifically on photoredox catalysis [3], where the chemical transformations are usually achieved by photons of the visible light spectrum in the presence of substoichiometric amounts of an absorbing sensitizer (photoredox catalyst). Photoredox catalysts are usually based on transition metal complexes (mostly Ru and Ir) and are generally very active; however, they are not suitable for industrial processes since they are toxic, harmful to the environment, and expensive. Organic dyes are preferable since they can be obtained at low cost and present good availability. Furthermore, they generally possess no toxic properties and are environmentally benign. Among the organic dyes, Eosin Y (EY) is one of the most popular due to its redox potential: its oxidation potential ranges between −1.06 and 1.10 V whereas its reduction potential ranges between +0.78 and +0.83 V (Figure 1a). The maximum of absorption sits at 539 nm and for this reason, green LEDs are usually employed as convenient light sources for Eosin Y activation (maximum emission around 530 nm) [4,5,6].

Upon excitation by light, Eosin Y undergoes a rapid intersystem crossing from the fundamental state to the lowest energy triplet state, which has a life time of 24 ms [7,8]. When in solution, Eosin Y equilibrates into four different forms due to the presence of two relatively acidic protons (pKa 2.0, 3.8 in water, Figure 1b). All these forms are pH dependent: at pH < 2, the protonated spirocyclic form **EY1** is in equilibrium with the neutral form **EY2**, whereas at 2 < pH > 3.8, the monoanionic form **EY3** is in equilibrium with the dianionic form **EY4,** which becomes predominant at pH > 3.8. Only **EY3** and **EY4** are catalytically active, but this behavior has generated a lack of clarity in many publications regarding the nature of the dye involved in the transformation [9]. In order to ensure the presence of the dianionic form **EY4**, Eosin Y sodium salt is employed as a photocatalyst, although consideration should be given to the experimental reaction conditions of the chemical transformation.

Eosin was largely used in photochemical reactions both as a homogeneous and heterogeneous catalyst after immobilization [10]. Immobilization of Eosin Y onto a solid support has many advantages, such as the possibility of recovery and reuse of the photocatalyst and the possibility of its use under flow conditions provided that the photocatalyst has a large surface area. We wish to report here our findings on the immobilization of Eosin Y onto Merrifield resin [11] and its application in the direct photochemical arylation of furan with aryldiazonium salts originally reported by König and coworkers in 2012 [12]. The synthesized supported photocatalyst was used in batch reactions under heterogeneous conditions, and its recovery and recycle were demonstrated for up to three times. The immobilized photocatalyst was then loaded in a packed-bed reactor and used under continuous flow conditions. Although the isolated yields of the flow reaction were lower than in batch, the productivity proved to be superior due to the short residence times.

## 2. Results and Discussion

### 2.1. Synthesis of Polystyrene-Supported Eosin Y

In Figure 2, a straightforward synthetic strategy for the immobilization of Eosin Y onto Merrifield resin is displayed. As a support, we chose high-loading Merrifield resin (1.2 mmol/g) 100–200 mesh. Merrifield resin is the most commonly used precursor polymer for solid-phase peptide synthesis. It is a terpolymer of styrene, with 4-vinylbenzylchlorid and 1–2% divinylbenzene as crosslinker. The resulting polymer beads from emulsion polymerization are macroscopic in size (Ø = 75 µm) and become, when swollen, a free-flowing powder sufficiently soft to not grind each other down when agitated. Eosin Y in its inactive hydrogen form is deprotonated by DIPEA and reacts in a nucleophilic pathway with the Merrifield resin. The reaction is reliable and reproducible and can be conducted on 20 g scale [11]. After 72 h at 80 °C in DMF as a solvent and DIPEA as a base, with mechanical stirring, the mixture was poured into a glass sintered funnel and extensively washed. The removal of unreacted Eosin Y from the solid material was ensured by extensive washing with several solvents (see the Experimental Section for details), as evidenced by the disappearance of the pink color and confirmed by TLC of the last washing solvent. This covalent approach for the preparation of solid supported catalyst should, in principle, give a more durable material. The maintenance of the morphological integrity of the support was confirmed by SEM analysis on MR-EY. The chemical composition of the new material was investigated by FT-IT, EDS, and elemental analysis (see the Appendix A for further details). Using this procedure, the loading of the material was 0.168 mmol/g, as determined by the weight difference and confirmed by elemental analysis.

The UV-vis spectrum of MR-EY is reported in Figure 1.

### 2.2. Application of Polystyrene-Supported Eosin Y in Photocatalysis

The prepared photocatalyst was tested in the direct arylation of furan **2** with diazonium salt **1a**, exploiting optimized König’s conditions [12]. A homemade custom-designed photoreactor [13] equipped with 530 nm light-emitting diodes (LEDs) (424 mW/cm^2^) was used as the irradiation source [14]. In the presented photoreactor, the LEDs are hermetically sealed inside a Pyrex glass tube to avoid glacier formation. The heat generation, which tends to burn high-power LEDs under sealed conditions, is counteracted by wrapping the LED strips around a central sublimator glass piece, which is water cooled [15]. For a detailed description and pictures of the photoreactor, see the Appendix A. The reaction vial was placed next to the photoreactor, with a distance between the vial and the LEDs < 1 cm, thus ensuring maximum exposition to light. The results of the direct arylation are illustrated in Table 1.

As expected, the reaction did not proceed in the absence of the photocatalyst (Table 1, entry 1). The use of 2 mol% homogeneous Eosin Y (non-supported) afforded 55% yield after a 24 h reaction (Table 1, entry 2). Further, 2 mol% MR-EY photocatalyst led to 44% yield after a 2 h reaction, which increased to 47% after a 4 h reaction and 53% after a 16 h and 24 h reaction (Table 1, entries 3–6). Increasing the amount of catalyst (5 and 10 mol%) led to an increase in the product yield (53% and 55% yield after 2 h, Table 1, entries 7–8). Increasing the ratio between the aryldiazonium salt **1a** and furan **2** from 1:10 to 1:20 led to the formation of the desired product **3a** in 75% yield after a 2 h reaction time (Table 1, entry 9). This result perfectly compared to that reported by König and coworkers (74% isolated yield after 2 h) [12]. Further increasing the **1a**:**2** ratio was not beneficial (Table 1, entry 10).

Having established the best reaction conditions, we next expanded the scope of the reaction using different aryldiazonium salts. The results are summarized in Table 2.

The aryl diazonium salts [21] were prepared according to the corresponding aniline availability in our laboratories [22]. Both electron-donating and electron-withdrawing groups on the phenyl ring can be used in the reaction. In our hands, no big difference in reactivity could be evidenced, in contrast with Professor König’s findings [12]. The desired products **3b-i** [12,16,17,18,19] could be isolated in satisfactory yields, ranging from 36% of the 2-Cl,4-NO_2_ derivative to 67% of the 4-NO_2_ compound (Table 2, entries 9 and 3). We would like to point out that no workup is necessary at the end of the reaction; once the LEDS are switched off, the reaction crude is separated from the catalyst by simple filtration and subjected to purification. It should be noticed that the obtained results are generally lower than those reported by König. As examples, aryl diazonium salt **1b** afforded compound **3b** in 40% yield while 54% yield was reported in the literature; aryl diazonium salt **1c** afforded compound **3c** in 67% yield vs. the 85% literature yield. This difference in reactivity could be ascribed to the presence of the solid support, which somehow lowers the reaction rate. However, the main advantage of the supported Eosin is its easy recovery by filtration and possible reuse in subsequent reactions and the possibility of running the reaction in a continuous flow, as illustrated below.

### 2.3. Recovery and Recycle and Flow Chemistry of Polystyrene-Supported Eosin Y

Once the performances of the supported Eosin were proved, we next investigated the stability of the photocatalyst in subsequent reactions between **1a** and **2**. At the end of the reaction, MR-EY was filtered onto a sintered glass funnel, copiously washed with dichloromethane and diethyl ether, and dried under vacuum until constant weight. The recovered MR-EY was reused under the same conditions. The reaction crude, containing the product and unreacted starting material only, was purified by column chromatography to afford pure **3a**. The results of these findings are summarized in Table 3. The reported yields are the average of three runs. In the first run, compound **3a** was recovered in 70% yield as reported in Table 1 and as reported by König and coworkers [12]. In the second and third run, the yield dropped to 45 and 40%, respectively. We then decided to explore the possibility of using MR-EY under continuous flow conditions.

A packed-bed reactor was then built using 1.23 g of MR-EY (*f* = 0.168 mmol/g; 0.207 mmol load inside the reactor) and 5 mm glass balls (for a detailed description, see SI). To ensure maximum irradiation, the packed-bed reactor was surrounded by three photoreactors equipped with green LED strips (424 mW/cm^2^) and a water-cooling system. A solution of aryldiazonium salt **1a** in DMSO and a solution of furan **2** in DMSO were fed by a syringe pump into the reactor with a certain flow rate and left to react for the desired residence time; the reaction crude was collected in a flask at the reactor output. The configuration of the system is depicted in Figure 3. Product **3a** was isolated in 56% yield after a residence time of 10 min (150 µL/min as the flow rate). After a residence time of 40 min and a flow rate of 38 µL/min, **3a** was isolated in 62% yield. A residence time of 10 min was then chosen for further investigation.

After these tests, the column was washed with 40 mL DMSO and freshly prepared solutions of aryldiazonium salt **1a** in DMSO and furan **2** in DMSO were fed into the reactor. The first three reaction volumes were discarded to allow the equilibration of the system, and then a sample of the crude reaction mixture was collected. Compound **3a** was isolated in 55% yield after a residence time of 10 min only (150 µL/min flow rate). The same packed-bed reactor was then washed again with DMSO and a solution of aryldiazonium salt **1b** in DMSO and furan **2** in DMSO were fed into the column. The first 3 reaction volumes were discarded, then a sample of the crude reaction mixture was collected to afford **3b** in a 42% isolated yield after a residence time of 10 min (150 µL/min flow rate). Finally, the whole procedure was repeated and aryldiazonium salt 1c was reacted with furan **2**, affording product **3c** in a 34% isolated yield. A comparison of the productivities of the different systems is summarized in Table 4. Although the isolated yields of the flow reactions are lower than the corresponding yields in batch, the productivity of the flow system is higher by a factor of at least 2.5. This increase in productivity is due to the very short residence time chosen for the flow reaction. This increase is further evidenced when considering the space-time-yield (STY), the metric typically used to compare reactors at different volumes. From batch to flow, a sharp increase in STY was observed: x9 for compound **3a**, x13 for compound **3b**, and x6 for compound **3c**. The sharp increase in the STY was obtained due to the shorter reaction time and the smaller reaction volume of the packed-bed reactor with respect to the batch. These findings imply that, when choosing a scale-up strategy, the packed-bed-reactor in this study is the more convenient and efficient choice because of the drastically intensified reaction conditions.

In conclusion, a straightforward and gram-scale synthesis of Eosin Y supported on Merrifield resin was developed. The supported photocatalyst was used in the direct C-H arylation of furan with differently substituted aryldiazonium salts, with yields comparable with those reported in the literature. The reaction workup was greatly simplified by the possibility of separating the immobilized photocatalyst by filtration and washing. Although the recycle of the catalyst was not completely efficient and satisfactory, the use of the supported Eosin Y in a packed-bed reactor allowed the productivity of the methodology to be greatly improved due to the short residence time. The flow system showed an increased productivity by a factor of at least 2.5. The sharp increase was even more evident with the STY, which proved to be higher by a factor of at least 6 in the case of the flow system with respect to the batch. We believe that the combination of supported photocatalysis and flow chemistry can contribute to the development of efficient and scalable synthetic methodologies.

## 3. Experimental Section

### 3.1. General Description of Reagents and Equipment

If not otherwise stated, the reagents, solvents, and such were used without further purifications. Eosin Y was bought from TCI chemicals and was either used without purification or was converted to sodium salt by the addition of aqueous sodium hydroxide solution. *N,N-*dimethylformamide was degassed prior to use. *N,N-*diisopropylethylamine was used without prior purification. Merrifield Resin High Loading 1.2 mmol/g was purchased from Merck. Acetonitrile was used in HPLC grade.

Reactions were monitored by thin-layer chromatography (TLC) on Macherey-Nagel pre-coated silica gel plates (0.25 mm) and visualized by UV light. Flash chromatography was performed on Merck silica gel (60, particle size: 0.040–0.063 mm). ^1^H NMR, ^13^C NMR, and ^19^F NMR spectra were recorded on a Bruker Avance-400 spectrometer in CDCl_3_ as solvents at room temperature. Chemical shifts for protons are reported using residual solvent protons (^1^H NMR: *δ* = 7.26 ppm for CDCl_3_) as the internal standard. Carbon spectra were referenced to the shift of the ^13^C signal of CDCl_3_ (*δ* = 77.0 ppm). The following abbreviations are used to indicate the multiplicity in the NMR spectra: s—singlet; d—doublet; t—triplet; q—quartet; dd—double doublet; ddd—doublet of doublet of doublets; dt—doublet of triplets; m—multiplet; quint—quintuplet; sext—sextuplet; sept—septet; br—broad signal; dq—doublet of quartets.

High-resolution mass spectra (HRMS) were acquired using a Bruker solariX XR Fourier transform ion cyclotron resonance mass spectrometer (Bruker Daltonik GmbH, Bremen, Germany) equipped with a 7 T refrigerated actively shielded superconducting magnet. The samples were ionized in positive ion mode using MALDI or ESI ionization sources.

ATR spectra were recorded using a Jasco FT/IR-4600 instrument. A KF-Technologies, NE-300 series Just Infusion syringe pump was employed for continuous flow applications.

### 3.2. Synthesis and Characterization of Solid-Supported Eosin Y Merrifield Resin (MR-EY)

In total, 10.0 g (8.33 mmol, 1.00 eq, *f* = 1.20 mmol/g) of Merrifield-Resin High-Load 100–200 mesh was introduced in a 250 mL 3-necked round-bottom flask. Then, 6.48 g (10.0 mmol, 1.20 eq) Eosin Y (hydrogen form) was added. The solid was mechanically stirred and 133 mL *N,N-*dimethylformamide was added, followed by 3.48 mL (20.0 mmol, 2.40 eq) diisopropylethylamine. After setting the temperature to 80 °C, the dispersion was stirred for exactly 72 h. After this time, the reaction mixture was poured into an oven-dried glass-sintered funnel (pore size 4, pre-weighed) and special care was taken to remove almost all the material out of the flask with generous amounts of methanol. The residue was infused with a mixture of water/THF/methanol and stirred with a glass rod. After infusing for 5 min, a vacuum was attached, and the washing liquid was filtered off. The vacuum was detached and the whole process was repeated 15 times. After this generous washing process, the process was repeated three times using dichloromethane. The washing flasks was changed and the remains in the funnel were dried by running a constant air stream through them for 5 h by attaching a vacuum. After this time, the filter was weighed again, and by the difference in weight, a preliminary catalyst loading was calculated, which amounted to *f* = 0.168 mmol/g. The catalyst appeared as a dark red solid. For the characterization of the heterogeneous photocatalyst, see the Appendix A.

### 3.3. General Procedure for the Direct C-H Arylation of Furan with Aryl Diazonium Salts under Batch Conditions

A 5 mL vial equipped with a magnetic stirring bar was loaded with MR-Eosin Y (0.02 equiv, 0.013 mmol, 77.3 mg), the desire aryl diazonium tetrafluoroborate (1 equiv, 0.662 mmol), and 2.6 mL dry DMSO. The resulting mixture was degassed by “pump-freeze-thaw” cycles (×2) via a syringe needle, then furan (20 equiv, 13.24 mmol, 970 µL) was added. The vial was attached with a rubber band to the wall of the photoreactor and irradiated using green LEDs for 2–24 h. After the desired time, the reaction mixture was quenched with water (1.5 mL) and filtered with a Millipore apparatus (0.1 m hydrophilic filter). MR-EY resin was recovered, and the crude mixture was then extracted with diethyl ether (5 × 10 mL) and the combined organic layers were dried over NaSO_4_, filtered, and concentrated under vacuum. Purification of the crude product was achieved by flash column chromatography using different mixtures of hexane:ethyl acetate as the eluent.

### 3.4. General Procedure for the Direct C-H Arylation of Furan with Aryl Diazonium Salts under Flow Conditions

In a typical experiment, syringe A was filled with a mixture obtained by dissolving 3.6 mmol of the desire aryl diazonium tetrafluoroborate in 10 mL of degassed DMSO (final concentration 0.36 M). Syringe B was loaded with a solution obtained by diluting 5.43 mL of furan in 4.57 mL of degassed DMSO (final concentration 7.42 M). Mixtures A and B were pumped into the packed-bed reactor at the desired flow rate using a syringe pump. Two reactor volumes were discarded before starting sample collection in order to achieve steady-state conditions. Different samples were collected according to the residence time into a vial containing 1.5 mL of water and protected by sunlight with aluminum foil. The crude mixture was then extracted with diethyl ether (5 × 10 mL) and the combined organic layers were dried over NaSO4, filtered, and concentrated under vacuum. Purification of the crude product was achieved by flash column chromatography using different mixtures of hexane: ethyl acetate as the eluent.

## Data Availability

Not applicable.

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
