# Peer review of "Supported Eosin Y as a Photocatalyst for C-H Arylation of Furan in Batch and Flow"

_molecules, 2022, doi:10.3390/molecules27165096_

Round 1
Reviewer 1 Report
The current manuscript presents results of a know reaction, that is the eosin Y-photocatalysed C-H arylation of furan with aryl diazonium salts, with the only difference that the eosin is covalently immobilized on a resin. The results appear not to be superior with respect to the homogeneous protocol, but the process has some advantages, e.g. the reuse of the photocatalyst. In addition performing the same reaction in a packed-bed reactor under continuous flow, a higher productivity has been achieved over the same period of time. For these reasons, the current reviewer considers this article as publishable in Molecules. I am wondering however, why there is limitation of the process on parent furan?
Author Response
We thank the Reviewer for her/his comments and we are happy to see that she/he considers our manuscript publishable in Molecules. Regarding the question, why there is limitation of the process on parent furan: we wanted to demonstrate that the covalently immobilized Eosin Y onto Merrifield resin is a useful heterogeneous photocatalyst for the C-H arylation of furan with aryl diazonium salts. Since the main advantage of a heterogeneous catalyst is its easy recovery and reuse, and the possibility of preparing a packed-bed reactor, we mainly focussed onto these two aspects. The parent furan was largely present in our laboratories, while substitued furans were not. For these reasons, we decided to investigate the reaction of furan, extending the scope to different aryldiazonium salts.

Reviewer 2 Report
Dear Authors:
This paper presented a very interesting report for Supported Eosin Y as a Photocatalyst for C-H Arylation of Furan in Batch and Flow. The overall structure and writing indeed require a significant improvement. I really would like to see this article in Molecules after major revisions.
1. In the original file, provide the full name of the synthesized compound instead of using the supported Eosin Y expression.
2. Add to the introduction the mechanism of action of eosin Y.
3. In the original file, in the Synthesis of Supported Eosin Y section, you did not provide all the information about the synthesis of MR-EY. You should provide the procedure in full so that readers can use it as a reference.
4. In the Synthesis of Supported Eosin Y section, you talked about the color change as a sign of the completion of the reaction. The color change is not a good indication of the completion of the synthesis. You should use TLC. Also, state the type and ratio of the slag used in TLC in the article.
5. Compare the absorption spectra of MR-EY and eosin Y in the same figure.
6. Synthesized compounds in Table 2, if known, should be referenced. But if they are unknown, only 1HNMR is not enough, and the 13CNMR and mass spectrum file should be added.
7. On page S7 of the Supporting Information file: DMSO-d6 is more correct. Please correct it.
Best Regards
Round 2
Reviewer 2 Report
Dear Authors
I read the revised manuscript (molecules-1835869). The authors have applied the requested corrections to the manuscript. In my opinion, the manuscript is now accepted for publication in Molecules.
Best Regards